# Ozone in Patients with Periodontitis: A Clinical and Microbiological Study

**DOI:** 10.3390/jcm11102946

**Published:** 2022-05-23

**Authors:** Ana Maria Ramirez-Peña, Arturo Sánchez-Pérez, Matilde Campos-Aranda, Francisco Javier Hidalgo-Tallón

**Affiliations:** 1Ministerio de Educación Superior Ciencia y Tecnología República Dominicana, Santo Domingo de Guzmán 10204, Dominican Republic; dra.aramirezp@gmail.com; 2Clínica Odontológica Universitaria, Hospital Morales Meseguer, Marques de los Vélez, 2° Floor, University of Murcia, 30008 Murcia, Spain; 3Escuela Universitaria De Osteopatía, Campus Universitario de Espinardo Edificio C, 2° Floor, 30100 Murcia, Spain; macampos@um.es; 4Department of Dermatology, Stomatology, Radiology and Physical Medicine, Universidad Católica San Antonio de Murcia (UCAM), Av. de los Jerónimos, 30107 Murcia, Spain; fjht63@gmail.com

**Keywords:** ozone, periodontitis, periodontal disease, periodontal debridement, nonsurgical periodontal therapy, biofilm

## Abstract

The purpose of this article was to assess the effectiveness of ozone therapy as an adjunct to mechanical therapy in periodontitis patients. Thirty-two patients diagnosed with generalized periodontitis were selected, with a total of 655 teeth examined. Each patient’s mouth was divided into four quadrants (the split-mouth model) to be randomly treated with four sessions of gaseous ozone or air. The following clinical variables were recorded: the gingival index, the periodontal clinical attachment loss, the Miller’s mobility index and the clinical improvements, as assessed through the visual analog scale (VAS). In addition, the microorganisms were qualitatively compared. After four weeks of treatment, the teeth of the ozone-treated quadrants showed statistically significant reductions in the gingival index and an improvement in the clinical attachment (*p* < 0.0001). The same treatment also significantly improved mobility by between 70% and 86% compared to the control group (*p* < 0.0001). Statistically significant differences were also recorded for the VAS (*p* < 0.0001). In the qualitative study of the subgingival flora, significant differences were observed (*p* < 0.0001). The overall results of this trial support the view that ozone treatment is effective and well tolerated in cases of generalized chronic periodontitis.

## 1. Introduction

Plaque-associated periodontitis is the most common form of periodontitis and is called “generalized periodontitis” when the attachment loss is present in more than 30% of the teeth [1]. The main etiological agents which cause periodontitis are the Gram-negative bacteria found in the subgingival plaque. It is not possible for the patient to suppress these pathogenic microorganisms due to their highly organized nature [2], which necessitates professional intervention through various strategies to reduce inflammatory activity and promote self-cleaning. The amount and virulence of the microorganisms and the host’s resistance mechanisms are crucial for the initiation and progression of periodontal impairment [3].

Different therapeutic modalities (surgical or nonsurgical) [4] are used to remove bacterial deposits and biofilms. Several studies have concluded that the complete removal of plaque by mechanical treatments in periodontal pockets measuring more than 5 mm is difficult; thus, it is advisable to use other methods to improve the effects of mechanical treatments [5,6,7].

In addition to mechanical control, the so-called chemical control of plaque is becoming increasingly widespread. The success of antimicrobials, both topical and systemic, is common in the current treatment of chronic diseases [8].

The application of ozone as an additional treatment represents a new approach in the management of periodontitis and may be considered a complementary treatment option [8]. It is considered to be an adjunctive therapy (i.e., a therapy given in addition to the main treatment to maximize its effectiveness). The effects of ozone on living organisms as an antimicrobial, analgesic, anti-inflammatory, immunostimulant (in terms of both cellular and humoral immunity), antihypoxic and a detoxifying agent have been published [9,10]. Ozone is recognized as a powerful germicidal agent against viruses, bacteria, fungi and spores and is able to kill all known types of Gram-positive and Gram-negative bacteria, including *Pseudomonas aeruginosa* and *Escherichia coli*, both of which are extremely resistant to antibiotics [11]. The broad-spectrum germicidal effects of ozone are due to its high oxidizing capacity, upon which typical mechanisms of microbial resistance have no effect [11]. Thus, after exposure to ozone, most microorganisms are disabled within 10 s. Anaerobic microorganisms are particularly sensitive in this respect, while *Candida* is the most resistant [12].

Ozone in dentistry can be administered in the form of gas, dissolved in water or can be oil-based [9] and has been proposed as an ideal therapeutic agent with few adverse effects. Moreover, the availability of ozone is almost unlimited, and it is inexpensive.

The intention of this work was to evaluate the effectiveness of insufflated medical ozone on periodontal pockets of patients with generalized chronic periodontitis.

Our null hypothesis is the absence of statistically significant differences between the use of medical ozone or atmospheric air as a treatment for periodontitis.

## 2. Materials and Methods

Thirty-two patients (ten men and twenty-two women), between 34 and 64 years of age, were selected from the Dental Clinic of Morales Meseguer Hospital in Murcia, Spain. The study was conducted from 21 June to 27 December 2021. To be included in the study, patients had to be diagnosed with generalized periodontitis, corresponding to stages II and III of the world classifications of 2018 [13]. Patients were recruited following the implementation of a nonprobabilistic sequential model.

The purpose of this study was explained to the patients, and informed consent was provided, read and signed. This consent was developed in accordance with the ethical principles for medical research in humans [14]. The Ethics Committee of the University of Murcia approved the study protocol with ID: 3399/2021. All patients received detailed information on the purpose of the study and were also informed of the possibility of withdrawing from the study at any time without explanation. This study conformed to Strengthening the Reporting of Observational Studies in Epidemiology (STROBE) guidelines.

Selected patients were required to have a minimum of 12 teeth evenly distributed in the four quadrants of the mouth. Patients with substance abuse (previous or ongoing), compromised immune systems, systemic diseases that could interfere with periodontal disease or hypersensitivity to ozone were excluded. Pregnant or lactating women were also not included. Of the participants, 56.25% were smokers, and 43.75% were nonsmokers.

The medical history of each subject had information on their concomitant pathologies and medication collected. Data relating to previous or ongoing treatments (pharmacological or not) for generalized chronic periodontitis were also collected. Initially, ultrasonic supragingival scaling was performed (Sonicflex 2003L de Kavo, KaVo Dental GmbHHead OfficeBismarckring 3988400 Biberach/Riß Germany), followed by polishing the surfaces with prophylactic paste and a brush. Each patient was instructed to continue their usual feeding and oral hygiene methods.

A week after performing the supragingival scaling, the first treatment session took place. Before applying the gases, the gingival index (GI), the periodontal clinical attachment level (CAL) and the Miller’s mobility index (MOV) were recorded and were measured again during both the third treatment session and three weeks afterwards.

At the same time, a qualitative examination of the subgingival flora was performed by taking samples from the 2 deepest pockets of each patient, which were evaluated by phase contrast microscopy (LED Optika B-350, OPTIKA S.r.l. Via Rigla, 3024010—Ponteranica (BG)—Italy). Intraoral photographs were also taken.

In both the third session and the third week of follow-up, patients were asked to mark on a visual analog scale (VAS) how they felt with respect to the disease. This scale allowed the intensity of pain to be measured, with maximum reproducibility among observers. The VAS consists of a horizontal line of 10 cm, at the ends of which are the extreme expressions of a symptom. The VAS is measured with a millimeter ruler. Intensity is expressed in millimeters.

In addition, during each visit after the first treatment session, the adverse reactions mentioned by each patient were recorded.

To improve the accuracy of our measurements, the data were always recorded by the same person, under the same conditions, always in the afternoon, and the explorer was calibrated for every five patients to the baseline measurements with a coincidence of 98% (the interclass correlation coefficient).

### 2.1. Ozone Insufflation Protocol

Using the split-mouth design, each patient’s mouth was divided into four quadrants, which were associated in pairs (first and third, and second and fourth) to be randomly treated (www.random.org (accessed on 1 May 2021)) with medical ozone or air (control) for four weeks (one application per week). The Quickly by Ozonline^®^ machine was used to obtain ozone (ECO3, Sevilla, Spain) (Figure 1). Depending on the randomization, freshly generated ozone (2 mL of ozone at a concentration of 30 µgr/mL) or 2 mL of air was insufflated into each periodontal pocket using a sterile needle (0.30 × 12 mm BL/LB) (Figure 2). A total of 655 teeth were treated. Of these, 333 were insufflated with ozone, and 322 were insufflated with air.

#### Subgingival Sampling and Microbiological Analysis

A qualitative examination of the subgingival flora was carried out, taking samples from the control group and the ozone group with the following procedure:

The supragingival plaque (above the gum line) was removed with a brush, and the tooth was insulated with sterile cotton rolls. With a periodontal curette, the subgingival plaque (below the gum line) was collected from the bottom of the periodontal pocket and transferred to a slide. The collected samples were emulsified in a drop of saline solution for evaluation by phase contrast microscopy. According to the qualitative determination of the flora, the patient was classified as healthy when cocci and bacilli were present, as a moderate risk when there was a predominance of bacilli, or as a high risk when there were spirochetes and other moving microorganisms (Table 1).

### 2.2. Statistical Methods

For the statistical analyses of the GI, CAL and MOV, descriptive statistics were used—essentially, the arithmetic mean and standard deviation. Confidence intervals for the means of both groups were also recorded at a confidence level of 95%, to determine whether there were significant differences between the two treatments.

To follow the evolution of the qualitative examination of the subgingival flora (ESF), a variance analysis for repeated measurements (ANOVA) and an analysis of the means were performed. In the case of patient-evaluated improvement (VAS), the mean and standard deviations for weeks 3 and 6 of treatment were used to ascertain significant differences.

The power of the study was evaluated posteriori at a value of 80.7%.

Finally, at a confidence level of 95%, estimates were made for the analysis of adverse reactions to determine how many patients suffered the same effects.

## 3. Results

No dropouts occurred during the study.

In the ozone-treated group, the GI, CAL and MOV showed significant reductions in the third and sixth weeks of the study, while for the control group (inflated air), no statistically significant differences were recorded in any of the variables studied.

Data for the GI pointed to favorable development, with a statistically significant improvement (*p* < 0.0001); the mean values changed from 1.37 at week zero to 0.57 at week three and 0.48 at week six. No significant changes were recorded in the control group during treatment or at week 6.

Regarding the periodontal CAL, a statistically significant improvement (*p* < 0.0001) was observed for the ozone-treated group, and the mean values changed from 4.05 at week 0 confidence interval CI 3.8–4.2) to 2.24 at week 3 and, finally, to 2.00 at week 6. No significant changes were recorded in the control group during treatment or at week 6.

For the ozone-treated group at week 0, the average mobility index was 0.38, while at week 3, the average dropped to 0.19, an improvement that persisted at week 6. In the control group, the average mobility values showed no significant changes throughout the study.

The improvement in tooth mobility at a confidence level of 95% was estimated to be between 70% and 86%. An analysis of the percentage improvement showed that more than 78% improved, with a *p* < 0.05 (Table 1).

Regarding the qualitative examination of the subgingival flora, there were significant differences between the ozone-treated group and the control group, with a variation in the type of microorganisms found in ozone-treated quadrants. This variation was numerically reflected with a range of 0–2, where 0 represents the healthy group and 2 represents the high-risk group. Table 2 illustrates these changes. In the ozone-treated group, the variation values significantly decreased between weeks 0 and 3 (*p* < 0.0001), while in the control group, the values showed no significant differences throughout the study.

The values of the VAS evaluated by the patients indicated a significant improvement between weeks 3 and 6 (*p* < 0.001).

An estimate of the percentage of adverse reactions at a confidence level of 95% showed that the percentage of patients with no adverse reactions was between 64% and 92%. Of the seven patients who had adverse reactions after ozone insufflation, one patient had bruxism, two patients had headaches and four patients experienced tenderness or tooth pain.

## 4. Discussion

Most studies on ozone therapy in periodontal disease have given positive results [15,16,17]. However, medical ozone, which is generated from medical oxygen, has not been used previously. We decided to apply this strategy since it is the most widely used in medicine, where the biological properties of ozone have been well demonstrated [18].

At the beginning of the study, the values of the ozone-treated quadrants and the control quadrants were similar, but throughout the study, a significant decrease in the recorded periodontal indices was observed for the ozone-treated quadrants, which can only be attributed to the use of medical ozone.

The GI showed a significant improvement from an initial average of 1.37 to a final value of 0.48 at the end of treatment. This response suggests that the gingival inflammation was slowed and must have been directly related to the anti-inflammatory effect of ozone.

Regarding the periodontal CAL, there was a statistically significant improvement in the quadrants treated with ozone, as reflected by a decrease in the probing depth. For this group, the values fell from an average of 4.05 mm before treatment to an average value of 2.00 mm at the end of treatment. This could be due to the disappearance of inflammation, a favorable regenerative process, and/or the formation of a long junctional epithelium. Similar degrees of improvement have only been recorded in treatments such as scaling and root planning [19,20,21] or periodontal surgery [22,23,24], which come with the drawbacks of requiring more time and being more invasive.

Ramzy et al. [16] and Kshitish and Laxman [24] have also published favorable results regarding the CAL and GI following irrigation by ozonized water (*p* < 0.001). Issac et al. [8] tested subgingival irrigation with ozonized water for 4 weeks in periodontal patients with probing depth > 6 mm. The probe depth showed a significant reduction after treatment (*p* < 0.01). In a similar study, Katti and Chava [2] used irrigated ozonized water and found that the mean probing depth values showed statistically significant differences between the 15th and 30th days of treatment (*p* < 0.05).

Dengizek et al. [25] compared the use of nonmedical gaseous ozone accompanied by scaling and root planing with scaling and root planing plus a placebo. After treatment, changes in the probing depth and the GI were similar in both groups, but there was a significantly higher increase in TGF-β levels in the ozone group than in the control group (*p* < 0.05).

Tooth mobility is another key parameter in periodontitis and predicts short-term tooth survival [26,27,28,29,30,31,32,33,34]. The proper treatment of tooth mobility is decisive [35,36,37,38,39,40], and for this purpose, different regenerative and splinting techniques have been developed, the effectiveness of which seems to lack a high level of evidence [36,38].

In the revised literature, we found no studies involving ozone therapy that evaluated tooth mobility. In our study, an improvement was evident in the ozone-treated quadrants compared with the air-treated quadrants. The average value at week 0 was 0.38, and this decreased to 0.14 at week 6. This fact may have been due to the disappearance of inflammation and swelling, and as ours was a very conservative treatment, it did not cut the remaining supracrestal fibers, whose reinsertion would be favored by respecting the collagen matrix of the cement.

A qualitative examination of the subgingival flora was also performed by observing the microorganisms through a phase contrast microscope. It was decided to use this simple technique because of the advantage of being able to perform the tests immediately, i.e., without the need for a laboratory to prepare the samples. However, it had the limitation of not being able to identify specific species. In our study, we recorded significant differences in the types of microorganisms found, with a predominance of nonpathogenic species in the ozone-treated quadrants.

Other authors have also analyzed microbiological parameters. For example, Issac et al. [8] assessed the effects of subgingival irrigation with ozonized water together with scaling and root planing compared with scaling and root planing alone. They found that the total counts of anaerobic colonies were significantly lower in the group treated with ozonized water.

Uraz et al. [7] used a polymerase chain reaction (PCR) and an enzyme-linked immunosorbent assay (ELISA) to analyze the crevicular fluid of patients with chronic periodontitis, in order to evaluate treatment with nonmedical gaseous ozone as an aid to scaling and root planing. With both techniques, there were significant reductions in the total bacterial counts, particularly for the scaling and root planing group a month after starting treatment and for both techniques at three months after treatment. In terms of species, only *Prevotella intermedia* was sensitive to ozone treatment in the third month of the follow-up.

In dentistry, due to the toxic effects of ozone by inhalation, the use of ozonized water or oils is preferred [8]. However, no feelings of discomfort or only minimal nasopharyngeal irritation were recorded during applications. Furthermore, as a safety measure, an aspirator was used simultaneously with insufflation. In addition, using a fine needle to insufflate the ozone allowed the gas to be applied with extreme precision to the periodontal pockets. In our study, the treatments were well tolerated, and few adverse effects were recorded (22%): one patient claimed to have developed bruxism, two patients had a headache and four patients experienced tenderness or toothache.

To standardize the type of patient, those with an established diagnosis of stages II and III periodontitis were chosen (i.e., those with formerly moderate chronic periodontitis). Although the percentage of smokers was high (56.25%), it was decided that this additional risk factor would be accepted, and it was found that the group of smokers showed significant improvements to the same extent as the nonsmokers. The percentage of patients who smoked was higher than the average for the Spanish population (56.25%). This result could potentially be explained by the fact that those seeking treatment in our university clinic were typically of a lower social class.

Prior to treatment with ozone or air, all patients underwent supragingival scaling so that subgingival plaques were not removed. In addition, patients were advised not to alter their diet and to continue with their usual oral hygiene practices. No mouthwash, brushes or toothpastes were prescribed, nor were the patients given oral hygiene instructions. In this way, we were able to evaluate the effectiveness of ozone without another variable interfering with the results.

In other studies that were similar to our own, participants were instructed in oral hygiene techniques [8,25].

Chlorhexidine has shown similar results in the treatment of periodontitis, but only after scaling and root planing [40]. However, chlorhexidine has disadvantages, such as a tendency to stain teeth and restorations and its toxicity to gingival fibroblasts, which can alter healing and cause flaking in the mucosa [37].

Kshitish and Laxman [24] conducted a study in patients diagnosed with generalized chronic periodontitis to evaluate and compare the effects of oral irrigation with ozonized water and 0.2% chlorhexidine. The authors observed a high percentage of reduction in the in the plaque index, the GI and the bleeding index in the group treated with ozonized water, compared to those treated with chlorhexidine irrigation. In addition, ozone treatment was more effective than chlorhexidine in reducing the amount of *Aggregatibacter actinomycetemcomitans*. Regarding the antifungal effect, ozonized water was very effective, while chlorhexidine showed no efficacy.

The mechanism by which ozone acts is not well established. As it has been mentioned, reactive oxygen species (ROS) generation could be responsible for the recruitment of neutrophils. However, the physiological purpose of this oxidative regulatory mechanism remains unknown [41].

As generalized chronic periodontitis is a periodontal disease with a relatively asymptomatic pathology, we decided to measure the degree of patient acceptance of the treatment, because this factor would indicate their willingness to adhere to the treatment. The VAS is easy to apply and was chosen to record the degree of patient satisfaction, despite its limitation of not being adapted to the split-mouth model. In our study, the values from the VAS decreased significantly between weeks 3 and 6, indicating that patients perceived the treatment as positive.

Finally, it should be noted that after the completion of the study, all patients received periodontal treatment free of charge at the Murcia University Dental Clinic.

Some limitations should be mentioned in our study: first, the studied population requested care in a university center due to the lower costs involved. Another limitation lies in the patients’ smoking habits, as the percentage of smokers in the study exceeded the average of Spain’s population. The follow-up period is also worth mentioning; although the response was short-term and very effective, we have no evidence of long-term results. Another limitation of our study is that any changes regarding hypoxia, metabolic acidosis or pH levels were not determined.

## 5. Conclusions

Within the limitations of this study, we can conclude that gaseous medical ozone is effective for the treatment of generalized periodontitis.

According to our results, the gain in the level of insertion, the decreased probing depth, the decreased plaque and mobility and the good degree of satisfaction indicate the clear benefits of using medical ozone as a minimally invasive, economic, painless and excellently tolerated therapeutic measure that is accepted by patients.

## Figures and Tables

**Figure 1 jcm-11-02946-f001:**
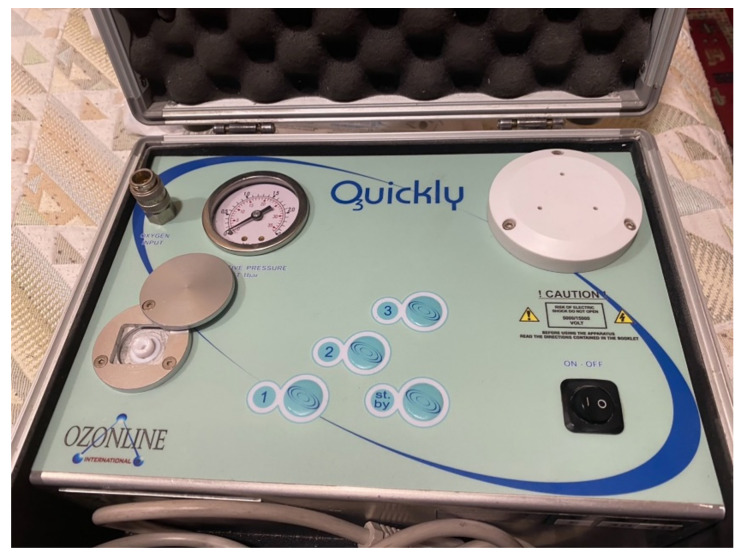
Device for the generation of ozone by transformation of medical oxygen (the Quickly by Ozonline^®^ (ECO3, Sevilla, Spain)).

**Figure 2 jcm-11-02946-f002:**
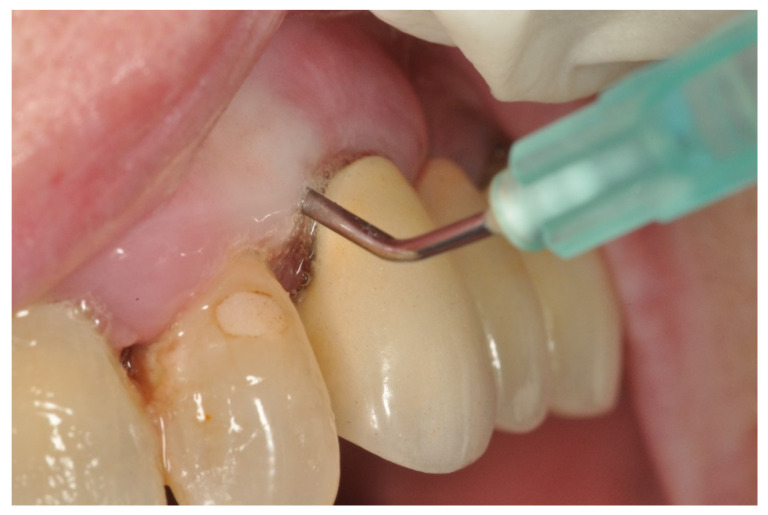
Administering ozone intrasulcularly through a cannula.

**Table 1 jcm-11-02946-t001:** Evolution of the gingival index, periodontal clinical attachment level and Miller’s mobility index.

Week	Treatment	GI		CAL		MOV	
		Mean	CI	Mean	CI	Mean	CI
0	Ozone	1.37	1.32–1.43	4.05	3.84–4.30	0.38	0.30–0.46
	Air	1.42	1.37–1.48	3.91	3.77–4.13	0.23	0.17–0.29
3	Ozone	0.57	0.52–0.62	2.24	2.07–2.43	0.19	0.12–0.26
	Air	1.47	1.33–1.43	3.99	3.8–4.19	0.21	0.15–0.27
6	Ozone	0.48	0.43–0.53	2.00	1.84–2.16	0.14	0.09–0.19
	Air	1.40	1.30–1.47	4.09	3.89–4.29	0.23	0.17–0.29

GI = gingival index, CAL = clinical attachment level, MOV = mobility, CI = confidence interval.

**Table 2 jcm-11-02946-t002:** Analysis of variance of the subgingival flora of the ozone group.

Origin of Variations	EFS	Sum of Type III Squares	Gl	F	*p*
Group 1. Ozone	Linear	17.016	1	212.321	0.0001
	Quadratic	5.672	1	117.627	0.0001
Error (G1)	Linear	2.484	31		
	Quadratic	1.495	31		

## Data Availability

Data from this study are available upon request to the corresponding author in anonymized excel format.

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
