# Peer review of "Ozone in Patients with Periodontitis: A Clinical and Microbiological Study"

_jcm, 2022, doi:10.3390/jcm11102946_

Round 1
Reviewer 1 Report
APPROPRIATENESS: The article entitled “Ozone in Patients with Periodontitis: A Clinical and microbiological Study” has a good appropriateness to the journal of Clinical Medicine. This article aims to evaluate the effectiveness of medical ozone insufflation in periodontal pocket. Interestingly, the results of this paper suggest that ozone treatment is effective and well tolerated in cases of generalized chronic periodontitis. CLARITY: This article is well written, and it is of interest in an international journal, however, some information is missing.
ORIGINALITY: The originality of the paper is good. In fact, I went through the current literature on the matter, and I found only a few articles on similar topics.
The authors need to perform the following major changes:
- Authors have pushed their work on the role of inflammation and its triggering factors, but no clinical applications are reported about this issue (see and discuss: Bressan, E., Ferroni, L., Gardin, C., Bellin, G., Sbricoli, L., Sivolella, S., Brunello, G., Schwartz-Arad, D., Mijiritsky, E., Penarrocha, M., Penarrocha, D., Taccioli, C., Tatullo, M., Piattelli, A., & Zavan, B. (2019). Metal Nanoparticles Released from Dental Implant Surfaces: Potential Contribution to Chronic Inflammation and Peri-Implant Bone Loss. Materials (Basel, Switzerland), 12(12), 2036.)
- Following tissue exposure to ozone, or other conditions such as hypoxia, metabolic acidosis may occur, shifting extracellular pH. On this basis, here such topic has been not properly reported.
- Please explain all the acronyms throughout the text
Minor:
- Main limitations should be reported
- Future prospects must be more related to clinical strategies: please improve this section accordingly, Please, improve the clinical application of ozone in regenerative procedures, comparing this approach to other differen ones, such as the use of nanovesicles to regenerate tissues
Author Response
Referee 1
- Authors have pushed their work on the role of inflammation and its triggering factors, but no clinical applications are reported about this issue (see and discuss: Bressan, E., Ferroni, L., Gardin, C., Bellin, G., Sbricoli, L., Sivolella, S., Brunello, G., Schwartz-Arad, D., Mijiritsky, E., Penarrocha, M., Penarrocha, D., Taccioli, C., Tatullo, M., Piattelli, A., & Zavan, B. (2019). Metal Nanoparticles Released from Dental Implant Surfaces: Potential Contribution to Chronic Inflammation and Peri-Implant Bone Loss. Materials (Basel, Switzerland), 12(12), 2036.)
The article recommended by the referee is of great interest and we have included it in our bibliography. In our study the treatment was performed on patients (in vivo) and in patients without implants (in periodontal patients), so it was not mentioned as a reference. However, its revision and incorporation represents a notable improvement in the study. Thank you for your suggestion
Reference 42: Bressan E, Ferroni L, Gardin C, Bellin G, Sbricoli L, Sivolella S, Brunello G, Schwartz-Arad D, Mijiritsky E, Penarrocha M, Penarrocha D, Taccioli C, Tatullo M, Piattelli A, Zavan B. Metal Nanoparticles Released from Dental Implant Surfaces: Potential Contribution to Chronic Inflammation and Peri-Implant Bone Loss. Materials (Basel). 2019 Jun 25;12(12):2036. doi: 10.3390/ma12122036. PMID: 31242601; PMCID: PMC6630980.
The following paragraph has been entered: “Reactive oxygen species (ROS) generation that could be responsible for the recruitment of neutrophils. However, the physiological purpose of this oxidative regulatory mechanism remains unknown.”
- Following tissue exposure to ozone, or other conditions such as hypoxia, metabolic acidosis may occur, shifting extracellular pH. On this basis, here such topic has been not properly reported.
We agree with the referee. However, in our work, ozone was applied topically, not injected into the tissues, but deposited at the bottom of the bag and in continuity with the surface of the teeth. Therefore, we assume that the metabolic changes suggested by the referee do not have an impact on the metabolic behavior in the tissue. Although ozone has a high capacity for penetration and diffusion, we believe that the amount present within the connective tissue is minimal.
We have included your comment as a limitation of our study. “Another limitation of our study is that neither hypoxia, nor metabolic acidosis or pH change has been determined.”
- Please explain all the acronyms throughout the text
All acronyms have been reviewed and explained before use
Minor:
- Main limitations should be reported
We have extended the limitations of the study
- Future prospects must be more related to clinical strategies: please improve this section accordingly, Please, improve the clinical application of ozone in regenerative procedures, comparing this approach to other different ones, such as the use of nanovesicles to regenerate tissues
Our work is based on the use of ozone as a coadjutant measure in cases of basic periodontal treatment. The referee's suggestion is interesting, but it exceeds the aims of our objectives, given that they are oriented towards a population that in general lacks the resources to assume regenerative treatments.
For this reason, we have included this limitation in our study. “Limitations: Some limitations should be mentioned in our study. First, the population under study. It is a population that requests care in a university centre due to its lower cost.”
Thank you for your time and effort dedicated to improving our work.
Reviewer 2 Report
Dear Authors,
After a careful analysis of the “Ozone in Patients with Periodontitis: A Clinical and Microbiological Study” manuscript, address the following aspects:
- Line 14: The first sentence requires a predicate.
- Please, state the null hypothesis.
- Was there a Power analysis performed?
- Lines 127, 128: “supragingival plaque” should be more correct.
- Table 3 is missing.
- Lines 211, 213, 217: “probing depth”.
- Line 248: “Prevotella intermedia”
- Line 250 requires a reference
- Lines 261-262: data on smokers/non-smokers was not presented in the Results section.
- Lines 263-266 should be moved into the Material and Methods section.
- Line 298: maybe “adjunctive treatment” should be more exact.
Author Response
Referee 2
- Line 14: The first sentence requires a predicate.
We have rewritten the sentence and changed it according to the predicate. “The purpose of this article was to assess the effectiveness of ozone therapy as an adjunct to mechanical therapy in periodontitis patients.” Thanks for the advice
- Please, state the null hypothesis.
We have included a null hypothesis. "Our null hypothesis is the absence of statistically significant differences between the use of medical ozone or atmospheric air as a treatment for periodontitis."
- Was there a Power analysis performed?
Since this was a pilot study, no a priori statistical power analysis was performed. We based our study on the central limit theorem (sample size 32 patients). However, posteriori we include the power obtained: The probability of a type II error (0.1926), as well as the statistical power, for this two-tailed test, for the given significance level of α = 0.05, and a sample size of n = 32. The critical value in this case is Zc = 1.96Z.
We obtain the power directly as: Power=1−β=1−0.1926=0.8074. We have included the power of the study in the statistical method (80.74)
- Lines 127, 128: “supragingival plaque” should be more correct.
We have corrected the term “supragingival plate” to be more specific. According to the International Journal of Dental and Health Sciences (IJDHS). The difference between the two types is the location of the plaque in relation to the edge of the gingival tissue. We have changed the term to "supragingival plate” by “supragingival plaque (above the gum line)." And “subgingival plate” by “subgingival plaque (below the gum line)”.
- Table 3 is missing.
We have renumbered the tables and corrected the missing table. Thank you.
- Lines 211, 213, 217: “probing depth”.
We have corrected "depth sounding". Thank you.
- Line 248: “Prevotella intermedia”
- intermedia has been replaced by Prevotella intermedia
- Line 250 requires a reference
We have included a reference to support the statement.
41 “Issac AV, Mathew JJ, Ambooken M, et al. Management of Chronic Periodontitis Using Subgingival Irrigation of Ozonized Water: A Clinical and Microbiological Study. J Clin Diagn Res. 2015;9(8): ZC29-ZC33. doi:10.7860/JCDR/2015/14464.6303”
- Lines 261-262: data on smokers/non-smokers was not presented in the Results section.
Indeed, the data of patients who were smokers were not included. Given the peculiarity of the sample (patients with low resources) most of them were smokers (56.25%) while the percentage of non-smoking patients was 43.75%. These percentages are higher than in the general Spanish population where the percentage of patients who currently smoke is below 30%. (By sex, the percentage of daily smokers was 27.6% in men and 18.6% in women). In addition, by occupational social class, a different behavior is observed in daily tobacco consumption according to sex. While in men there is a clear social gradient from 20.0% in the upper class to 33.6% in the lower class, in women there is no clear pattern. We have included a reference to this data in the discussion. "The percentage of patients who smoked was higher than the average for the Spanish population (56.25%), possibly influenced by the less favoured social class seeking treatment in our university clinic".
- Lines 263-266 should be moved into the Material and Methods section.
We agree with the referee and have moved lines to the Material and methods section.
- Line 298: maybe “adjunctive treatment” should be more exact.
We have specified the term adjunctive treatment according to the National Institute of Health (NIH) definition including its definition "Adjunctive therapy is therapy given in addition to the main treatment to maximize its effectiveness."
Thank you for your time and effort dedicated to improving our work.
Round 2
Reviewer 1 Report
none to add
Author Response
Referee 1
Comments and Suggestions for Authors:
None to add
Answers:
Thank you for your comments and for your helping hand in improving this work.
Yours sincerely,

Reviewer 2 Report
Line 34: ”explored areas” is very vague. Please, update the information by focusing on the new Classification system (regarding the 30% threshold of involved teeth).
Lines 224, 226, 230: I think there might be a misunderstanding. The correct term is ”probing”, not ”sounding”. Therefore, it would be more appropriate to be referred as ”probing depths” (as the actual value found upon probing”. Also, ”deep pockets” was just correct.
Line 303: please, prefer ”As it has been mentioned..”
Lines 316-321: Please, prefer: ”Some limitations should be mentioned in our study. First, the studied population requested care in a university center due to lower costs involved. Another limitation lies in the smoking habit, with a percentage of smokers that exceeds the prevalence in the general population of Spain. The follow-up period is also worth mentioning;although the response is short-term and very effective, we have no evidence of long-term results.”
Line 345: Please, remove ”References”
Author Response
Referee 2
Comments and Suggestions for Authors:
Line 34: ”explored areas” is very vague. Please, update the information by focusing on the new Classification system (regarding the 30% threshold of involved teeth).
Lines 224, 226, 230: I think there might be a misunderstanding. The correct term is ”probing”, not ”sounding”. Therefore, it would be more appropriate to be referred as ”probing depths” (as the actual value found upon probing”. Also, ”deep pockets” was just correct.
Line 303: please, prefer ”As it has been mentioned..”
Lines 316-321: Please, prefer: ”Some limitations should be mentioned in our study. First, the studied population requested care in a university center due to lower costs involved. Another limitation lies in the smoking habit, with a percentage of smokers that exceeds the prevalence in the general population of Spain. The follow-up period is also worth mentioning; although the response is short-term and very effective, we have no evidence of long-term results.”
Line 345: Please, remove” References”
Answers:
Line 34: We have replaced “explored areas” according to the new classification system by
“When the attachment loss is present in more than 30% of the teeth”.
Lines 224, 226 and 230: We agree with the referee and have replaced all the terms "sounding deep" by "probing deep". Thank you.
Line 303: We have corrected the sentence “As has been mentioned” and replace by.” As it has been mentioned”
Lines 316-321: We appreciate the correction and have accepted the suggestion. The limitations are now worded according to the suggestion as: ”Some limitations should be mentioned in our study. First, the studied population requested care in a university center due to lower costs involved. Another limitation lies in the smoking habit, with a percentage of smokers that exceeds the prevalence in the general population of Spain. The follow-up period is also worth mentioning; although the response is short-term and very effective, we have no evidence of long-term results.” Thank you.
Line 345: We have erased the word ”References”
We hope that this revised version will now be judged ready for publication in the Journal of Clinical Medicine, and look forward to hearing from you.
Yours sincerely,